# Two-Species Forests at the Treeline of Siberian Mountains: An Ecophysiological Perspective under Climate Change

**DOI:** 10.3390/plants10040763

**Published:** 2021-04-13

**Authors:** Nina Pakharkova, Anna Kazantseva, Ruslan Sharafutdinov, Irina Borisova, Vladimir Gavrikov

**Affiliations:** Institute of Ecology and Geography, Siberian Federal University, 660041 Krasnoyarsk, Russia; nina.pakharkova@yandex.ru (N.P.); anna74083@gmail.com (A.K.); ruslanate@mail.ru (R.S.); irina_borisova77@mail.ru (I.B.)

**Keywords:** treeline, chlorophyll, carotenoids, fluorescence, *Pinus*, *Abies*

## Abstract

In an underexplored region of the East Sayan mountains, ecophysiological traits of two conifers, *Pinus sibirica* Du Tour and *Abies sibírica* Ledeb., have been studied. The goal was to predict which of the species co-dominating the same habitat is more vulnerable under prospective climate change. Along a transect from the treeline to the floodplain, photosynthetic pigment content and electron-transport rate (ETR) were measured in needles of neighboring trees of the species. From 570 to 1240 m a.s.l., *P. sibirica* does not suffer from stress factors during the growing season, while *A. sibirica* does. The latter is reflected in a decrease of pigment content and ETR with the increase of altitude. A stronger climate-change trend (probably to more dry and warm conditions) will likely favor the shift of *P. sibirica* upper in altitudes, and only under the pine shelter might the fir survive the changes.

## 1. Introduction

Alpine and subalpine forests are among the most vulnerable biomes because they occupy the very tolerance edges of areas of species [1,2]. Although effective remote-sensing methods have been developed to monitor alpine treeline ecotones [3,4], they could face large uncertainties in attempts to project dynamics of a particular forest under climate change. To perform such projections, one has to measure key trees’ traits in situ. A shift of treeline downslope seems to be strongly connected to cold events lasting for centuries [5]. However, a reverse upslope treeline movement may not automatically follow climate warming because the movement depends on the survival of seedlings outside the canopy, where they are subjected to freezing events and extreme light intensities, as was shown for Smith fir (*Abies georgei* var. *smithii*) [6]. Other species, like *Larix gmelinii* (Rupr.) in the Putorana Mountains [7], showing abundant regeneration ensure a significant upslope shift of the treeline during the last century.

To move to upper altitudes or retreat to lower elevations will largely depend on the tree’s ability to tolerate the stressing factors of the tolerance edges, more so under climate change. Multispecies forests present interesting objects because different tree species may display divergent reactions to changing environments and therefore lay a basis for a future change in species compositions. Guo et al. [8,9] compared stem radial growth parameters of spruce and fir genera at the alpine treeline of the Tibetan Plateau. Through comparing the radial increment series, it was found that higher growing-season temperatures may be beneficial for spruces, but can induce drought stress on firs.

In the mountains of southern Siberia, radial and apical increments of conifer saplings (*Larix sibirica* Ledeb., *Pinus sibirica* Du Tour, and *Abies sibirica* Ledeb.) at the treeline ecotone have been studied [10,11]. During the last three decades, *P. sibirica*, *L. sibirica,* and *A. sibirica* saplings showed an increase in increments, both radial and apical, as well as invasion into the tundra zone. It has been estimated that a raise in summer mean temperatures by 1 °C drives the pine saplings 150 m up in altitude. The larch increment correlated with summer temperatures, while the increment of the pine and fir correlated with both temperatures and precipitation. According to the authors’ estimations, under the current raise of summer temperatures, larch moves upslope by 1 m of elevation per decade.

Pakharkova et al. [12,13] used another approach to foresee how two-species mountain forests may alter their dominant species composition and the treeline position in the West Sayan ridge. The authors measured the content of photosynthetic pigments and the rate of electron transport in needles of codominating *Pinus sibirica* and *Abies sibirica*. The measurements were performed along a transect from a closed forest stand to the treeline and up into the tundra zone. The forests grew under an obvious climate change, with the yearly mean temperatures definitely rising, while the amount of precipitation remained about the same. The photosynthetic apparatus of *P. sibirica* seemed to be more adapted to harsh high-altitude conditions, e.g., excess UV light, because the content of chlorophylls decreases faster in *P. sibirica* with altitude than in *A. sibirica*. The same is valid for the electron-transport rate, which decreased at higher altitudes faster in *P. sibirica* than in *A. sibirica*. These adaptations may preserve the needles from UV light damage. In addition, *P. sibirica* can retain moderate needle temperatures during daytime sunshine hours, while the temperature of *A. sibirica* needles is on average higher (24.2 vs. 43.3 °C on sunny days). In higher altitudes, *A. sibirica* needles suffered from burn injuries and dried up, which prevented the fir saplings from colonization of habitats above the treeline. The *P. sibirica* trees and saplings compensated for the UV damage through a relative growth of carotenoids in the pool of pigments and effective transpiration, so the pine saplings could invade the habitats outside the treeline.

For a better understanding of the mechanisms of plant acclimation to environmental conditions, the processes of photosynthesis and fluorescence of chlorophyll are often studied. This approach is based on the fact that light energy absorbed by chlorophyll *a* molecules in the photosystem II (PSII) has three possible further paths: it is (i) used by the PSII reaction center (RC) during photosynthesis, (ii) dissipated as heat, or (iii) emitted as light with a shifted wavelength (ChlF). These paths correlate with plant health, as well as other ecophysiological factors affecting plants [14,15].

Chlorophyll fluorescence (ChlF) analysis has provided a wealth of information on the physiology of conifers, especially regarding plant responses to various environmental factors. Chlorophyll fluorescence measurements are usually very sensitive, and various instruments and analytical methods have been developed that can be used both individually and globally [15,16,17,18].

In this study, we continue to extend the approach used in previous studies [12,13] over a different geographic area. The goal of the study was to project the possible alteration at the treeline ecotone in a rather remote and heavily underexplored area of East Sayan under current climate change. To do this, it was necessary to establish relationships between *P. sibirica* and *A. sibirica,* which codominate most of the area, and to foresee the reaction of the species to the long-term trend in regional climate. If species at their upper limit experience stress, this stress could be detected in fluorescence and the needle pigment content. Hence, we hypothesize that some physiological indicators (such as the maximum quantum yield, electron-transport rate, chlorophyll, and carotenoid content) will change with the deviation of environmental factors from an optimal level.

## 2. Results

### 2.1. Chlorophyll Fluorescence

The rate of electron transport in needle cells (ETR) was taken as one of the indicators of photosynthetic activity. As evidenced by the data presented in Figure 1, during the active growing season (early August), the Siberian pine showed insignificant fluctuations in the speed of electron transport within altitudes from 570 to 1240 m a.s.l. With an increase in altitude, the speed of electron transport in one- and two-year-old needles in the pine had statistically significant differences (Appendix A, Table A1), while this indicator decreased significantly in two-year-old fir needles (Figure 1).

This suggests that the photosynthetic apparatus of *P. sibirica* functions stably and is not affected by any limiting factors at these altitudes. In addition, the observation may be due to species differences concerning the light saturation point of photosynthesis.

It was observed that one-year-old fir needles in fir had a lower speed of electron transport compared to two-year-old needles.

In *P. sibirica*, a light-saturation point was located at a higher light-intensity level compared to *A. sibirica*, which is seen in the light curves (Figure 2). A sharp decrease of ETR in fir needles gave evidence that the photosynthesis was inhibited under the treeline conditions (Appendix A, Table A1).

The maximum quantum yield of the PSII (Fv/Fm) significantly differed in needles of the conifer species at the highest and lowest points of the transect (Figure 3). At the lowest point, the value of Fv/Fm in the fir needles was higher than that of pine needles, while the relation was the opposite at the highest point (Appendix A, Table A1). Thus, the maximum quantum yield also gave evidence that photosynthesis was inhibited in fir at the treeline level.

### 2.2. Pigments

The chlorophyll content in pine needles slightly increased with altitude; on the contrary, a slight decrease was observed in fir (Figure 4).

With an increase of altitude from 682 to 1240 m, the average chlorophyll content in the one- and two-year-old needles of *P. sibirica* increased (not significant) by 1.14 times. In the needles of *A. sibirica*, on the other hand, the chlorophyll content decreased by 0.94 times in the one-year-old (not significant) and 0.72 times in two-year-old needles (significant). Significant differences in the change of chlorophyll content with increasing altitude were found only for two-year-old fir needles (Appendix A, Table A2).

## 3. Discussion

The soil data we obtained in the sampling points corresponded in general to our previous experience regarding the distribution of soil properties in mountainous forest lands. Though we did not perform a special test of how the soil characteristics were distributed, our data (not shown) suggested that the soil conditions may be rather even over the sampling area.

At different stages of the growing season, various environmental factors can limit the photosynthetic activity of both Siberian pine and fir. At the beginning of the growing season, the critical conditions of treeline habitats are a combination of negative soil temperatures, low positive air temperatures, and high illumination, aggravated by the reflection of light from spots of the snow cover. All this leads to burns and drying out of needles, which are especially pronounced in fir and prevent the advancement of its seedlings above the treeline [13]. Siberian pine partially compensates for these factors by increasing the content of carotenoids in needles and by later resumption of photosynthetic activity [12], which allows Siberian pine seedlings to occupy habitats above the forest boundary.

Data have been reported showing that fir needles in *Abies pinsapo* Boiss. and *Abies alba* Mill. growing under high light intensity in the undergrowth of plantations had different traits in anatomy and photosynthetic functions [19]. However, the authors admitted that the role of such changes in ultrastructural needle anatomy in explaining the response of mesophyll to the light environment has not been demonstrated in field conditions.

In the summertime, when the temperature is not limiting for these species, other factors (humidity, illumination, etc.) can limit their growth. Siberian pine and Siberian fir have different ecological requirements (ecological valence) for these factors. The appearance and vitality of woody plants at the upper limit of their growth reflect the regime and dynamics of the influence of external factors on one hand, and the response of the organism on the other.

The light-driven reactions of photosynthesis have to fit downstream biochemical processes, and photosynthesis follows a light saturation curve. At very low light intensity, oxygen consumption by respiration overcomes oxygen evolution by photosynthesis, until the compensation point is reached. At increasing light intensity, the photosynthesis rate, measured as photosynthetic oxygen evolution, increases linearly with the light, because in this phase light is the limiting factor. When light irradiance overcomes the rate of downstream biochemical reactions, the excess absorbed energy is dissipated via nonphotochemical energy quenching, and photosynthetic oxygen evolution reaches a plateau. Therefore, light energy applied in excess concerning the saturation level of photosynthetic electron transport does not contribute to biomass accumulation, but is dissipated as fluorescence and heat [20].

A slight increase in the content of chlorophyll in the pine needles is probably associated with a decrease in the length of the growing season. In fact, the data gathered closest to the research area weather station (WMO 295,760 Ujar) evidenced that the mean summer monthly temperatures (May through August) showed either a flat tendency or even a decrease from 1989 to 2020.

Siberian pine, as a light-loving plant, makes the most of all the incoming energy, as evidenced by the increase in the content of carotenoids (Figure 5, Table A2). It is known that the main functions of carotenoids are to protect chlorophyll molecules from irreversible photooxidation and to absorb light as additional pigments [21,22].

Carotenoids, which are part of the chlorophyll–protein light-absorbing complexes (antenna), perform the function of “collectors” of light in the summer period. In the mountains, however, the conifers show a significantly different adaptation at the beginning of the growing season—an increase in the content of carotenoids in needles indicates compensatory mechanisms for an excess of solar radiation under low temperatures. Thermal dissipation of excess absorbed energy, at the antenna level, is a fast and efficient protective strategy [23]. In addition, the differences in morphological and leaf attributes due to acclimation to each particular environment rely on different photoprotective mechanisms [24].

A decrease of chlorophyll and carotenoid contents in fir needles with an increase in altitude, more open forest stands, and closeness to the upper treeline indicates a complete light saturation of photosynthesis under these conditions. At the same time, a decrease in the speed of electronic transport gives evidence of photoinhibition of photosynthesis.

Thus, one can see that fir, as a dark coniferous species, cannot occupy open habitats; the only possible way for the colonization of fir seedlings above the treeline under possible climate change is to “follow” the *P. sibirica* trees.

## 4. Materials and Methods

### 4.1. Area and Study Objects

Kuturchin Belogoriye (“belogoriye” ≈ white mountains) is a mountain ridge, about 80 km long within the Eastern Sayan, with a maximum altitude of 1876 m. About 80% of the rocks are composed of the mixed-grained biotite leucogranites massif, while syenites and granosientes are less common. The age of the mountain massif determined by different methods is within the range of 450–490 million years.

The climate of the region is sharply continental, with long and severe winters and cool summers with unstable weather, during which most of the precipitation falls. At altitudes of 900–1300 m, the average January temperature ranges from −25 to −17 °C, and from 12 to 14 °C in July. The distribution of precipitation is closely related to the orientation of the mountain slopes: on the western and southwestern slopes, open toward humid air currents, precipitation of up to 800 mm or more falls per year, and up to 430 mm falls in the northern foothills. The maximum temperature ranges from 6 °C in January to 38 °C in July. At an altitude of more than 1600 m, snowfields persist throughout the summer, but their area has greatly decreased in recent years.

The main types of landscapes of the region are mountain-taiga and alpine stony tundra. Typical mountain-taiga landscapes, occupying more than 60% of the territory, are developed on the slopes of all main ridges and in river valleys. Dark coniferous (spruce, Siberian pine, fir) forests prevail. The upper treeline runs at an altitude of 1450–1550 m.

The largest rivers in the study area are the Mana and Mina, the watershed of which is the ridge. The Mina River is snow-fed with many swamps in its valley. Here, typical species are sedges, green mosses, and sphagnum mosses, and among the woods Scotch pine and birch are common. The area is poorly affected by economic activity and heavily understudied from a scientific viewpoint; an emerging sector is ecological tourism. Historically, from the early 1830s until the 1970s, loose gold was mined in small streams that are tributaries of the Mina River.

The studies were performed within a forest catena set up on a north-faced slope of the ridge (Figure 6). The catena is limited by the GPS coordinates: 54°51′33.80″ N, 94°16′7.91″ E; 54°48′33.94″ N, 94°15′53.80″ E (Figure 7). Within the catena, four sampling points were chosen based on visual considerations of facies types found. The reason to set up a sample point was therefore that the points fell within different facies to cover the facies (and soil) variability. Vegetative shoots (without generative organs) were collected from lower thirds of crowns of young trees standing at the forest stand edge, which ensured good light conditions.

The sampling and measurements were performed in mid-April 2020, which was the end of the growing season in the area. Thus, the methods slightly differed from those applied in a previous study [13], in which sampling was done at the very beginning of the growing season.

### 4.2. Soil Conditions

Along the entire study transect, the soil conditions were recorded. They presented a picture typical for most mountainous slopes of moderate angles of the gradient. That is, the habitats developed from eluvial facie at the treeline through trans-eluvial facies along the slope to super-trans-aqual facie at the hill bottom. The soil types according to World Reference Base (WRB-2014) are given as follows:1200 m a.s.l, Haplic Folic Fractic Skeletic Ferralic CAMBISOLS Arenic Densic;1147 m a.s.l, Leptic Folic Greyzemic Cambic Skeletic PHAEOZEMS Densic Abruptic Loamic;1049 m a.s.l., Sceletic Fractic Umbric LEPTOSOLS Densic Humic Loamic;890 m a.s.l., Leptic Folic Greyzemic Cambic Skeletic PHAEOZEMS Densic Abruptic Loamic (4);682 m a.s.l., Gleyic Floatic Skeletic HISTOSOLS Loamic Petrogleyic.

### 4.3. Measurements of Pigment Content

The sampling of trees was performed in the following way. At each sampling point, three healthy young fir and pine trees were chosen at the edge of the forest stand. The sizes of the trees varied, but were between 4 and 7 m in height. At the lowermost sampling point, only *P. sibirica* specimens were found, thus they were sampled regardless of the absence of *A. sibirica*.

To determine the pigment composition in needles, the first-year (5 samples per tree) and second-year growth (5 samples per tree) peripheral shoots at the bottom of the crown were sampled.

The content of chlorophylls was quantified on a CCM-200 plus Chlorophyll Content Meter (Hoddesdon, Herts, UK). The CCM-200 plus provides a programmable measurement-averaging option.

The content of carotenoids was quantified in an alcohol extract on a SPEKOL 1300 Analytik Jenna AG spectrophotometer (Jena, Germany). The cut shoots were delivered to the laboratory in dense paper bags within a day, and then the shoots were placed in vessels with water and kept in the same light and temperature conditions. The calculations were carried out according to the formulas of Smith and Benitez [25].

### 4.4. Measurement of Needle Fluorescence

Chlorophyll fluorescence (ChlF) can measure the efficiency of the Photosystem II (PSII), which is in charge of the light depending on reactions in photosynthesis [26]. When the amount of light absorbed exceeds the amount of energy that the plant can utilize, the efficiency of PSII is reduced, resulting in a delay of photosynthesis that affects the CO_2_ uptake. Measuring ChlF is a straightforward method for detecting evidence of photodamage and evaluating seedling stress caused by excessive absorbed light energy [27]. ChlF has been suggested as a good estimator for seedling stress and possible performance in the field [28,29].

Induced chlorophyll fluorescence was measured using a JUNIOR-PAM handheld fluorometer. The electron-transport rate (ETR) coefficient was taken as the main indicator, μmol of electrons m^−2^·s^−1^, calculated through the WinControl-3 software (Heinz Walz GmbH, Pfullingen, Germany). The electron-transport rate (µmol electrons/(m^2^·s) was derived from the Y(II) and PAR (photosynthetic active radiation, µmol photons m^−2^·s^−1^). The data represent factory-adjusted values that correspond to the PAR at a 1 mm distance from the tip of a 400 mm JUNIOR-PAM fiber-optic sensor.

The maximum quantum yield of the PSII (Fv/Fm) also was measured: Fv/Fm = (Fm − Fo)/Fm, which is the maximum photochemical quantum yield of photosystem II. Fo is the basic fluorescence yield (relative units) recorded with low measuring-light intensities; Fm is the maximal chlorophyll fluorescence yield when photosystem II reaction centers are closed by a strong light pulse (relative units).

Because chlorophyll fluorescence is a measure of re-emitted light (in the red wavebands) from PSII, any ambient light can interfere with the measurement of fluorescence, and thus many early systems had to be used in darkness and/or highly controlled light environments. This issue was overcome by the development of modulating systems in which the light used to induce fluorescence (the measuring beam) is applied at a known frequency (modulated), and the detector is set to measure at the same frequency [30]. In this way, the detector will only measure fluorescence that results from excitation by the measuring beam, and will not permit interference from ambient light. The clear advantage of this is that measurements can be made without darkening the leaf.

### 4.5. Statistical Treatment of Data

For the statistical treatment of the data, the ANOVA approach was applied (two-factor with replication and one-factor with replication). For two-factor ANOVA, the following factor pairs were tested: needle age and altitude, PAR (brightness of flash of PAM-fluorimeter) and species, and needle age and species. For one-factor ANOVA, the following factors were tested: altitude and species. Any tests for homoscedasticity and normality of the data were not performed due to a low amount of data. Pearson analysis was used to check for linear relationship between variables. All the calculations were performed with the Data Analysis add-in of Microsoft Excel.

## 5. Conclusions

In the context of the currently observed rapid climate change [31], the ranges of some conifer species do not shift from south to north, but are narrowed due to a rapid shift in the southern limits. The northern limit is moving more slowly, which is associated with the difficulties of “seizing” new territory. The same tendencies are observed in the zone of altitudinal zonation, sometimes even more clearly.

In this case, as evidenced by the data obtained both at the beginning of the growing season [13] and in the second half of it, and presented in this article, *P. sibirica* had more successful adaptive mechanisms for moving higher above the treeline compared to *A. sibirica*.

At the beginning of the growing season at the upper treeline, the critical conditions were a combination of negative soil temperatures, positive air temperatures, and high illumination. By the end of the growing season, a decrease in the photosynthetic activity of fir needles at higher altitudes can be caused by excessive solar irradiation of open habitats combined with a decrease in the concentration of chlorophyll and carotenoids in the needles. *A. sibirica*, as a dark coniferous species, needs shading of seedlings and young plants, so it will only move above the treeline under the canopy of other trees. Under the given environmental conditions, this “pioneer” species is likely to be *P. sibirica,* thanks to its physiological and ecological adaptive traits

## Figures and Tables

**Figure 1 plants-10-00763-f001:**
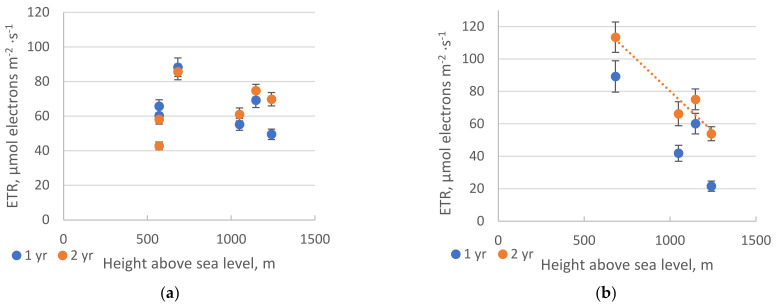
Electron–transport rate (ETR) in one– and two–year–old needles of *P. sibirica* (**a**) and *A. sibirica* (**b**). The significant slope is shown on the graph. Pearson analysis revealed significant linear regression in *A. sibirica* for two–year–old needles (*p* < 0.05). The vertical bars represent the standard deviation. From each sample point, three measurements of both one– and two–year old needles were taken. Here and further “yr” = year.

**Figure 2 plants-10-00763-f002:**
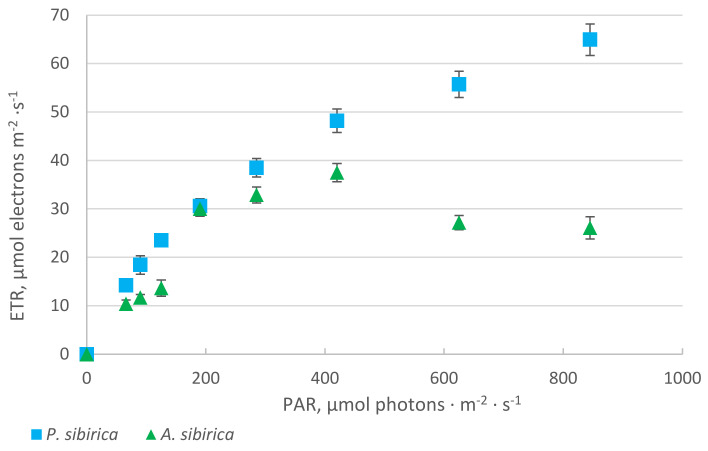
Change of ETR in needles of *A. sibirica* and *P. sibirica* against PAR. The vertical bars represent the standard deviation. For each PAR level, three measurements for two–year old needles were done. PAR = photosynthetic active radiation.

**Figure 3 plants-10-00763-f003:**
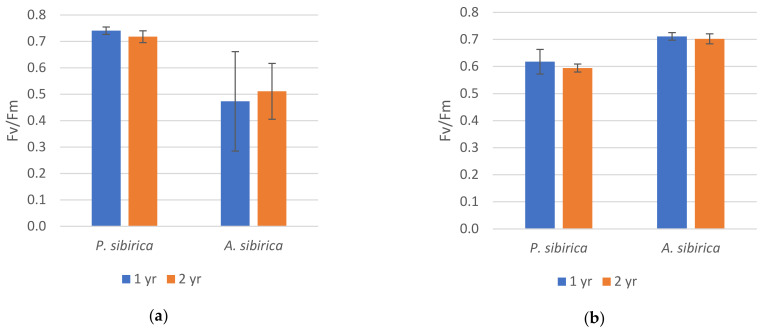
The maximum quantum yield of the PSII (Fv/Fm) in *P. sibirica* and *A. sibirica* needles at the highest point (**a**) and the lowest point (**b**) of the study transect. The vertical bars represent the standard deviation. At each sample point, three measurements of one– and two–year–old needles were taken for both *P. sibirica* and *A. sibirica*.

**Figure 4 plants-10-00763-f004:**
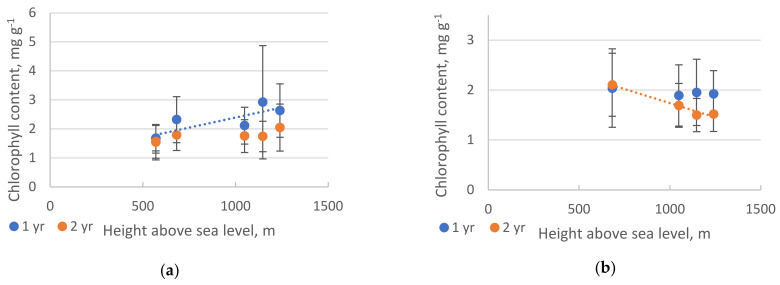
Chlorophyll content (dry weight) in needles of *P. sibirica* (**a**) and *A. sibirica* (**b**). The significant slopes are shown on the graph. Pearson analysis revealed significant linear regression in *P. sibirica* for one–year–old needles and in *A. sibirica* for two–year–old needles (*p* < 0.05). The vertical bars represent the standard deviation. At each sample point, 15 measurements of both one– and two–year–old needles were taken.

**Figure 5 plants-10-00763-f005:**
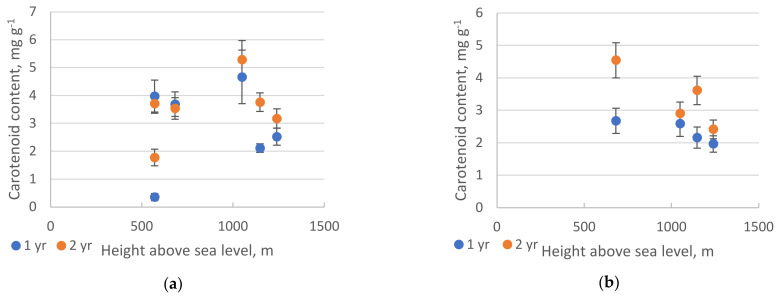
Content of carotenoids (dry weight) in needles of *P. sibirica* (**a**) and *A. sibirica* (**b**). Pearson analysis did not reveal any significant linear regression in either species (*p* > 0.05). The vertical bars represent the standard deviation. At each sample point, four measurements of both one– and two–year–old needles were taken.

**Figure 6 plants-10-00763-f006:**
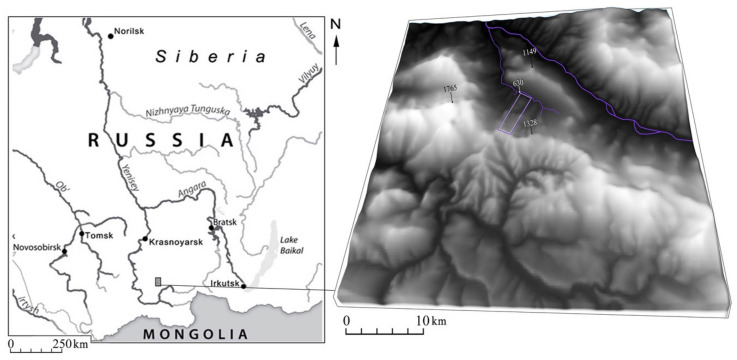
The geographical location of the study area.

**Figure 7 plants-10-00763-f007:**
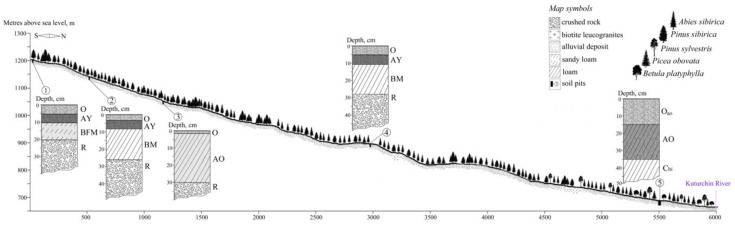
Profile of the forest catena in which the sampling was performed.

## Data Availability

The data presented in this study are available on request from the first author.

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
