# Peer review of "Two-Species Forests at the Treeline of Siberian Mountains: An Ecophysiological Perspective under Climate Change"

_plants, 2021, doi:10.3390/plants10040763_

Round 1
Reviewer 1 Report
The current version of the manuscript, albeit improved, still has several major shortcomings.
There are still major sections in the results that should be moved to the discussion. If separating results and discussion, as is the case in this manuscript, the authors need to keep the results to summary statements pertaining to the data, not their interpretation. For example, 128-145 is more appropriate for the discussion.
The authors need to make a stronger case for their research approach. For example, I am missing a clear hypothesis statement (e.g., If species at their upper limit experience stress, this stress could be detected in fluorescence. Hence we hypothesize that…”).
The analysis and presentation require further edits. Significant differences should be indicated in the figures, for example through the use of letters or asterisks. When testing whether a trend is evident, trendlines should be drawn (e.g., Figure 4), at least where the trends is significant.
It is not clear to me why P. sibirica has 5 data points compared to 4 data points for A. sibirica (Figs. 1, 4, 5). In the methods, it is mentioned that there were four sampling sites (218), not 5.
The implementation of the soil data is still insufficient. I am not saying that soil data are irrelevant, but the authors cannot claim that sampling soil was “necessary in order to take into account the possible influence of soil factors on the vital state of plants, if these differences are very significant” if they didn’t test this. Unless the actual soil data (i.e., numerical values) are included in the analysis in order to explain PS2 or chlorophyll differences, no claims of “significance” can be made. The soil data as is can merely be used in a speculative framework.
Lastly, I recommend proofreading the manuscript for spelling and grammar issues before resubmission.
Author Response
Thank you so much for your comments.
Q: There are still major sections in the results that should be moved to the discussion. If separating results and discussion, as is the case in this manuscript, the authors need to keep the results to summary statements pertaining to the data, not their interpretation. For example, 128-145 is more appropriate for the discussion.
A: The text fragment has been transferred to a relevant place in Discussion (L. 169-184).
Q: The authors need to make a stronger case for their research approach. For example, I am missing a clear hypothesis statement (e.g., If species at their upper limit experience stress, this stress could be detected in fluorescence. Hence we hypothesize that…”).
A: The following hypothesis has been inserted at the end of the Introduction (L. 86-89):
If species at their upper limit experience stress, this stress could be detected in fluorescence and the needle pigment content. Hence, we hypothesize that some physiological indicators (such as the maximum quantum yield, electron transport rate, chlorophyll, and carotenoid content) will change with the deviation of environmental factors from an optimal level.
Q: The analysis and presentation require further edits. Significant differences should be indicated in the figures, for example through the use of letters or asterisks. When testing whether a trend is evident, trendlines should be drawn (e.g., Figure 4), at least where the trends is significant.
A: We believe that a further adding of details into the figures is unnecessary. The dispersion analysis that we used did not perform pairwise comparisons of samples, thus the focus on these would be a confusing thing. Similarly, no tests of trend significance were done (like in regression analysis), so the trends will detract the attention and make the figures a bit cumbersome.
Q: It is not clear to me why P. sibirica has 5 data points compared to 4 data points for A. sibirica (Figs. 1, 4, 5). In the methods, it is mentioned that there were four sampling sites (218), not 5.
A: At a lower habitat, only P. sibirica specimens were found, so we decided to sample them nevertheless. An indication of this was added into the Materials and Methods section (L. 245-246).
Q: The implementation of the soil data is still insufficient. I am not saying that soil data are irrelevant, but the authors cannot claim that sampling soil was “necessary in order to take into account the possible influence of soil factors on the vital state of plants, if these differences are very significant” if they didn’t test this. Unless the actual soil data (i.e., numerical values) are included in the analysis in order to explain PS2 or chlorophyll differences, no claims of “significance” can be made. The soil data as is can merely be used in a speculative framework.
A: As we did not perform a special test regarding the soil conditions, we added a correspondent explanation that we hypothesized a minor influence of soil conditions (L. 136-139).
Q: Lastly, I recommend proofreading the manuscript for spelling and grammar issues before resubmission.
A: The manuscript was spellchecked through automated software.
Reviewer 2 Report
I would suggest to integrate the work with the comparative analysis of trends in ETR and ETP and in climate variables (e.i., monthly rainfall, monthly mean temperature ....).
Row 39. '... ability to withstand the stressing factors ...'. Perhaps using 'tolerate' in place of 'withstand'?
Row 83. ' ... East Sayan under ongoing climate ...'. Maybe 'under current climate ...'?
Author Response
Thank you so much for your comments.
Q: I would suggest to integrate the work with the comparative analysis of trends in ETR and ETP and in climate variables (e.i., monthly rainfall, monthly mean temperature ....).
A: Yes, this might be an interesting research direction. Unfortunately, as we mentioned, the area is a heavily underexplored region. No weather records are known for the area. The closest weather station is located at a distance of ca. 100 km and its information should be used with caution.
Q: Row 39. '... ability to withstand the stressing factors ...'. Perhaps using 'tolerate' in place of 'withstand'?
A: The sentence has been corrected as suggested.
Q: Row 83. ' ... East Sayan under ongoing climate ...'. Maybe 'under current climate ...'?
A: The sentence has been corrected as suggested.
Round 2
Reviewer 1 Report
Upon further review, I agree with the authors decision to omit indication in the figures. However, if the authors did not perform a type of linear model/regression for the data in Figure 4, the authors cannot proclaim “increases” or “decreases”, even if using “slightly” in their results. If no significance is detected (or no model is run to test for significance), any trends are purely subjective.
Regarding the discussion on soil: much better. Please change phrasing in lines 139-140 from “we hypothesize and took as a premise that” to “our data (not shown) suggest that”
Author Response
Dear Reviewer,
thank you for your comments.
Q: Upon further review, I agree with the authors decision to omit indication in the figures. However, if the authors did not perform a type of linear model/regression for the data in Figure 4, the authors cannot proclaim “increases” or “decreases”, even if using “slightly” in their results. If no significance is detected (or no model is run to test for significance), any trends are purely subjective.
A: Significance of the slopes in Figs 1,4,5 has been added to the captions of the Figs with correspondent p-values indicating the significance. The references to the significance issues have been added to the relevant places in the text.
Q: Regarding the discussion on soil: much better. Please change phrasing in lines 139-140 from “we hypothesize and took as a premise that” to “our data (not shown) suggest that”
A: The phrase has been changed as suggested.
This manuscript is a resubmission of an earlier submission. The following is a list of the peer review reports and author responses from that submission.
Round 1
Reviewer 1 Report
The paper wants to discuss the development of the co-dominance of two tree line species (Pinus sibirica Du Tour. and Аbies sibírica Ledeb.) in Siberia under climate change. It finds clear differences in ecophysiological traits between the species, which show that Pinus siberica is better adapted to the high PAR levels related to high altitudes and open canopy than Abies siberica. This implies that Pinus siberica will be the winner of the two species, when it comes to inhabiting new habitats above the treeline as may happen under climate change. The manuscript is short and compact, which I like.
The outcome as stated at the end of the last paragraph, is however not very well stated in the abstract, which implies that Abies siberica might be vulnerable to the higher temperatures under a changing climate in the treeline ecotone, for which the data shows no indication whatsoever. Where both species build the treeline at the moment Abies sibirica might even profit more than Pinus sibirica from climate change and warmer temperatures, since Abies expresses higher maximum quantum yield of the PSII at lower altitudes (warmer temperatures). This statement is made under the assumption that the climate at the treeline towards the end of the century, will be more like the climate at lower altitudes at the moment, where Abies performs better.
So please make clear what you want to (and more important can) show with this paper not only in the abstract but also at the end of the introduction and analog make a strong statement at the end of the discussion (which is already quite clear but could be better).
I did not find out at which time of the vegetation season you took your samples (especially since you start the discussion with differences between stages of vegetation season)
I further did not find out how many trees were sampled at one site, please add this to the methods part and also to the figures and further add error bars.
I would prefer if the colors (orange and blue) you chose for your figures, would not mean something different on every figure, maybe better use symbols to differentiate between species in figure 2.
Please state were the samples for figure 2 were taken (lowest site or highest site or all sites?).
You don’t discuss the possible implication of the different soil types, on different sites, could you add a sentence to the discussion?
Reviewer 2 Report
In their manuscript “Two-species forests at tree-line of Siberian mountains: an ecophysiological perspective under climate change”, the authors present chlorophyll content and fluorescence parameters of two co-dominant conifers species along an elevational gradient. The manuscript as it stands right now is not ready for publication due to major shortcomings:
- The methods are not explained in sufficient detail. For example, sizes are not mentioned. Statistical analyses are not mentioned/explained.
- It is not clear how this manuscript is different from Pakharkova, N.V.; Borisova, I.V.; Sharafutdinov, R.A.; Gavrikov, V.L. Photosynthetic Pigments in Siberian Pine and Fir under 319 Climate Warming and Shift of the Timberline. Forests 2020, 11, 63; doi:10.3390/f11010063. More importantly, the current manuscript lacks depths to stand by itself.
- The overall organization of the manuscript need considerable edits. The introduction lacks any mentioning of chlorophyll parameters and how they are important today. Instead, there is a misplaced paragraph on this topic in the results (90-100). The results also include several sections that should be discussions instead (e.g, 85-88, 125-129, 135-142).
- The figures require considerable edits. No error bars are shown, no trendlines are given. Why is Figure 3 a bar diagram and not similar to all other elevation gradient figures?
- Several sections include irrelevant information, or at least information/data that is never used in the analysis (e.g. 191-197, 211-221).
- Lastly, but probably most importantly, is that the conclusion drawn by the authors is questionable at best. This is an empirical study on one fluorescence parameters but the authors treat it as such that it clearly explains all species movement in this ecosystem. This needs to be relativized.